# Towards Zero-shot Relation Extraction in Web Mining:
# A Multimodal Approach with Relative XML Path

**Zilong Wang**
University of California, San Diego
zlwang@ucsd.edu

**Jingbo Shang**[*]
University of California, San Diego
jshang@ucsd.edu

## Abstract

The rapid growth of web pages and the increasing complexity of their structure poses a challenge for web mining models. Web mining models are required to understand semi-structured web pages, particularly when little is known about the subject or template of a new page. Current methods migrate language models to web mining by embedding the XML source code into the transformer or encoding the rendered layout with graph neural networks. However, these approaches do not take into account the relationships between text nodes *within* and *across* pages. In this paper, we propose a new approach, ReXMiner, for zero-shot relation extraction in web mining. ReXMiner encodes the shortest relative paths in the Document Object Model (DOM) tree of the web page which is a more accurate and efficient signal for key-value pair extraction within a web page. It also incorporates the popularity of each text node by counting the occurrence of the same text node across different web pages. We use contrastive learning to address the issue of sparsity in relation extraction. Extensive experiments on public benchmarks show that our method, ReXMiner, outperforms the state-of-the-art baselines in the task of zero-shot relation extraction in web mining.

## 1 Introduction

The internet is a vast repository of semi-structured web pages that are characterized by the use of HTML/XML markup language. Compared to plain text in traditional natural language understanding tasks, these web pages possess additional multimodal features such as the semi-structured visual and layout elements from the HTML/XML source code. These features can be effectively generalized across different websites and provide a richer understanding of the web pages (Lockard et al., 2018, 2019, 2020).

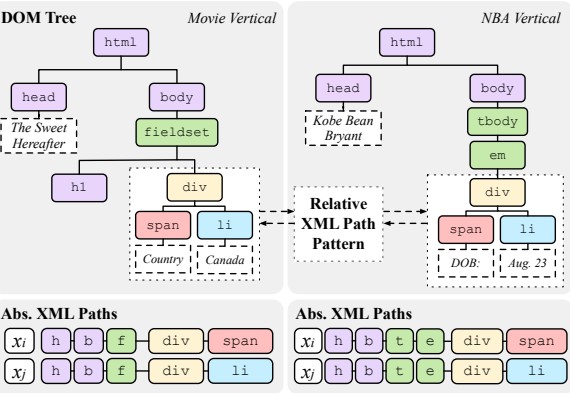

Figure 1: The structural information from semi-structured web pages. Based on the DOM Tree from the HTML source code, the absolute and relative XML Paths are extracted. We believe the web page structure is well modeled by the XML Paths to predict the attribute of text nodes and the relative XML Paths provide extra signals to predict the relation between text nodes.

The dynamic nature of the modern internet poses significant challenges for web mining models due to its rapid pace of updates. It is infeasible to annotate emerging web pages and train targeted models for them. Modern web mining models are expected to perform zero-shot information extraction tasks with little prior knowledge of emerging subjects or templates (Lockard et al., 2020; Chen et al., 2021). In this context, the multimodal features extracted from the HTML/XML source code as well as the textual contents are crucial for dealing with zero-shot information extraction task on the countless emerging web pages.

Previous approaches to the problem of zero-shot web mining have primarily focused on creating rich representations through large-scale multimodal pre-training, utilizing XML Paths of text nodes[1] (Lin et al., 2020; Zhou et al., 2021; Li et al., 2022b). As shown in Figure 1, XML Paths are sequences of tags (e.g., div, span, li) indicating the location of

---

∗ Jingbo Shang is the corresponding author.

[1] https://en.wikipedia.org/wiki/XPath

the text node in the DOM Tree[2] of the page. These pre-training approaches extend vanilla language models by embedding the absolute XML Paths but fail to take into account the relative local relationship expressed by the relative XML Paths. The related nodes tend to be close to each other in the DOM tree, which results in a long common prefix in their XML Paths, as shown in Figure 1. Such local relation is more common than the absolute XML Path patterns. Therefore, it is easy to transfer the relative XML Paths to new web pages, and the relative XML Paths serve as a more efficient and meaningful signal in predicting the relation between text nodes.

Additionally, existing web mining approaches tend to treat each web page separately and focus on memorizing their various templates, ignoring the fact that the relevance across different web pages of the same website is also meaningful to identify the related text nodes (Zhou et al., 2021; Li et al., 2022b; Lockard et al., 2020). Intuitively, a text node is more likely to be a key word if it appears frequently in a collection of web pages and its surrounding words are not fixed. For example, in web pages about NBA players, the statistics about the height, age are common text fields in the player introduction, so the text nodes, such as "Height:" and "Age:" should appear more frequently than other text nodes and the surrounding text contents should be different.

In light of the aforementioned challenges in web mining, we propose a web mining model with Relative XML Path, ReXMiner, for tackling the zero-shot relation extraction task from semi-structured web pages. Our approach aims to learn the local relationship *within* each web page by exploiting the potential of the DOM Tree. Specifically, we extract the shortest path between text nodes in the DOM Tree as the relative XML Path, which removes the common prefix in the XML Paths. Inspired by the relative position embedding in T5 (Raffel et al., 2020), we then embed the relative XML Paths as attention bias terms in the multi-layered Transformer. Additionally, we incorporate the popularity of each text node by counting the number of times it occurs *across* different web pages, and embed the occurrence logarithmically in the embedding layer. Furthermore, we address the data sparsity issues in the relation extraction task by

adopting contrastive learning during training which is widely used in related works (Su et al., 2021; Hogan et al., 2022; Li et al., 2022a). We randomly generate negative cases and restrict their ratio to the positive ones, allowing the model to properly discriminate related node pairs from others.

By learning from the relationships between text nodes *within* and *across* pages, ReXMiner is able to effectively transfer knowledge learned from existing web pages to new ones. We validate our approach on web pages from three different verticals from the SWDE dataset (Hao et al., 2011), including Movie, University, and NBA. The relation labels are annotated by Lockard et al. (2019). We summarize our contribution as follows.

- We propose a novel multimodal framework, ReXMiner, that effectively exploit the relative local relationship *within* each web page and incorporate the popularity of text nodes *across* different web pages in the relation extraction task.
- We represent the relative local relation and the popularity of text nodes in the language models through relative XML Paths in the DOM Tree and the occurrence number of text nodes across different web pages.
- Extensive experiments on three different verticals from SWDE dataset demonstrate the effectiveness of ReXMiner in the zero-shot relation extraction task in web mining.

**Reproducibility.** The code will be released on Github.[3].

## 2 Related Work

**Information Extraction in Web Mining** How to efficiently and automatically gathering essential information from the internet is always a hot topic in the academia of natural language processing and data mining due to the enormous scale and vast knowledge within the internet. The open information extraction task in web mining is originally proposed by Etzioni et al. (2008) and further developed by following works, including Fader et al. (2011); Bronzi et al. (2013); Mausam (2016) which rely on the syntactic constraints or heuristic approaches to identify relation patterns, and Cui et al. (2018); Lockard et al. (2018, 2019); Xie et al. (2021); Li et al. (2023) which introduce neural networks to solve the task under supervision or distant supervision settings. Our proposed method follows the task formulation of the zero-

---

[2]https://en.wikipedia.org/wiki/Document_Object_Model

[3]github.com/zlwang-cs/ReXMiner-release

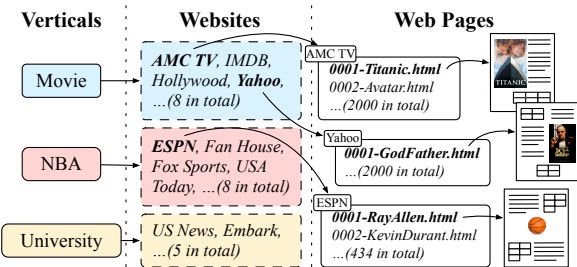

Figure 2: The web pages in the SWDE dataset. There are three verticals, Movie, NBA, University. Each vertical includes several websites. Each website includes hundreds web pages.

shot relation extraction in web mining proposed by ZeroShotCeres (Lockard et al., 2020) where the models are required to transfer relation knowledge from the existing verticals to the unseen ones. ZeroShotCeres adopts the graph neural network to understand the textual contents and model the layout structure. It finally produces rich multimodal representation for each text node and conduct binary classification to extract related pairs.

**Layout-aware Multimodal Transformers** The pre-trained language models, such as BERT (Kenton and Toutanova, 2019), XLNet (Yang et al., 2019), GPT (Brown et al., 2020), T5 (Raffel et al., 2020), are revolutionary in the academia of natural language processing. It achieves state-of-the-art performance in text-only tasks. To further deal with multimodal scenarios, various features are extracted and incorporated into the Transformer framework. Recent study has shown that it is beneficial to incorporate multimodal features, such as bounding box coordinates and image features,into pre-trained language models to enhance overall performance in understanding visually-rich documents (Xu et al., 2020, 2021; Huang et al., 2022; Gu et al., 2021; Wang et al., 2022b). Similarly, web pages are rendered with HTML/XML markup language and also represent layout-rich structures. Multimodal features from the DOM Tree or rendered web page images are incorporated in the pre-trained language models to solve the tasks in the semi-structured web pages (Lin et al., 2020; Zhou et al., 2021; Li et al., 2022b; Wang et al., 2022a).

## 3 Problem Formulation

The zero-shot relation extraction in web mining is to learn knowledge of related pairs in the existing web pages and transfer the knowledge to the unseen

ones (Lockard et al., 2020). The unseen web pages should be orthogonal to the existing ones with regard to vertical, topic, and template. The zero-shot setting requires the web mining models to extract relevant pairs based on both the textual content and the DOM Tree structure of web pages. Specifically, each web page is denoted as a sequence of text nodes, $P = [x_1, x_2, ..., x_n]$, where $n$ is the number of nodes in the page. Each node involves textual contents and the XML Path extracted from the DOM Tree, $x_i = (w_i, xpath_i)$. The goal of the zero-shot relation extraction task is to train a model using related pairs, $(x_i \rightarrow x_j)$, from a set of web pages, and subsequently extract related pairs from unseen ones. For example, as shown in Figure 2, one of our tasks is to train models with web pages from Movie and NBA verticals and test the models with web pages from the University vertical.

## 4 Methodology

We extend the text-only language models with multimodal features and propose a novel framework, ReXMiner, for zero-shot relation extraction task in web mining. Figure 3 shows the components in our framework. We adopt the absolute XML Path embedding in MarkupLM (Li et al., 2022b), and further extend it with popularity embedding and relative XML Path attention. To cope with the sparsity issue in the relation extraction task, we adopt the contrastive learning strategy where we conduct negative sampling to control the ratio between positive cases and negative cases.

### 4.1 Absolute XML Path Embedding

We follow the idea in MarkupLM and embed the absolute XML Paths in the embedding layer. We introduce it in this section for self-contained purpose. The XML Path is a sequence of tags from HTML/XML markup language (e.g., div, span, li). Both of the tag names and the order of tags are important to the final representation. Therefore, in the embedding layer, we first embed each tag as a embedding vector, and all these tag embeddings are concatenated. To be more specific, we pad or truncate the XPath to a tag sequence of fixed length, $[t_1, ..., t_n]$, and embed the tags as $\text{Emb}(t_1), ..., \text{Emb}(t_n)$ where $t_i$ is the $i$-th tag and $\text{Emb}(t_i) \in \mathbb{R}^s$ is its embedding. We further concatenate the vectors as $\text{Emb}(t_1) \circ ... \circ \text{Emb}(t_n) \in \mathbb{R}^{n \cdot s}$ to explicitly encode the ordering information, where $\circ$ is the operation of vector concatenation.

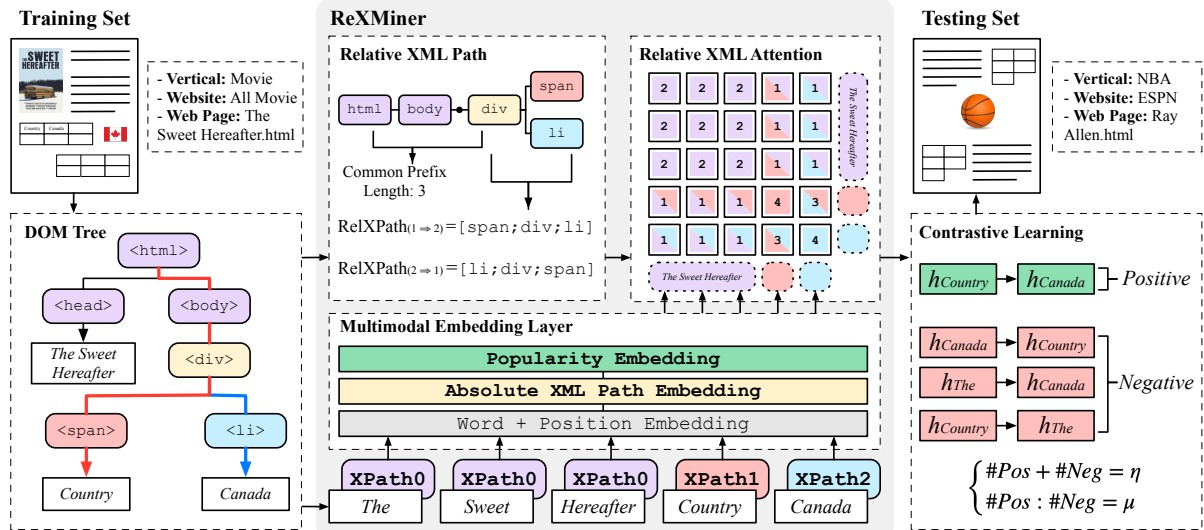

Figure 3: The framework of ReXMiner. We extract the DOM Tree of each web page from the HTML source code and further extract the absolute and relative XML Paths. We embed the popularity of text nodes and absolute XML Paths in the embedding layer and embed the relative XML Paths in the attention layers. We reduce the binary classification loss of the relation pairs sampled by negative sampling. In this figure, we train ReXMiner using web pages from the Movie vertical and test it on unseen web pages from the NBA vertical.

To fit in with the hyperspace of other embedding layers $\in \mathbb{R}^d$, a linear layer is used to convert the concatenation into the right dimension.

$$\text{AbsXPathEmb}(xpath_i)$$
$$=\text{Proj}(\text{Emb}(t_1) \circ ... \circ \text{Emb}(t_n)) \in \mathbb{R}^d$$

where $\text{Proj}(\cdot)$ is a linear layer with parameters $W \in \mathbb{R}^{ns \times d}$ and $b \in \mathbb{R}^d$.

## 4.2 Popularity Embedding

We propose Popularity Embedding to incorporate the occurrence of the text nodes into the pre-trained framework. Web pages from the same website use similar templates. The popularity of a certain text node across different web pages of the same website is meaningful in the relation extraction task in the web mining. Intuitively, a text node is more likely to be a key word if it appears frequently and its neighboring words are not fixed.

In details, given a text node $(w, xpath)$ and $N$ web pages $P_1, ..., P_N$ from the same website, we iterate through all the text nodes in each web page and compare their textual contents with $w$, regardless of their XML Paths. We count the web pages that involves nodes with the same text and define the number of these web pages as the *popularity* of $w$. Thus, higher popularity of a text node means that the same textual contents appears more fre-

quently in the group of web pages.

$$\sigma(w, P) = \begin{cases} 1, & \text{if } \exists xpath', \text{s.t.}(w, xpath') \in P \\ 0, & \text{otherwise} \end{cases}$$

$$pop(w) = \sum_{i=1}^{N} \sigma(w, P_i)$$

where $pop(w)$ is the popularity of $w$. Then we normalize it logarithmically and convert the value into indices ranging from 0 to $\tau$. Each index corresponds to an embedding vector.

$$\text{PopEmb}(w) = \text{Emb}\left(\left\lfloor \tau \cdot \frac{\log pop(w)}{\log N} \right\rfloor\right) \in \mathbb{R}^d$$

where $\text{Emb}(\cdot)$ is the embedding function; $\tau$ is the total number of popularity embeddings; $d$ is the dimension of embedding layers.

Formally, along with the absolute XML Path embedding, the embedding of the $i$-th text node, $(w_i, xpath_i)$, is as follows.

$$e_i = \text{PopEmb}(w_i) + \text{AbsXPathEmb}(xpath_i)$$
$$+ \text{WordEmb}(w_i) + \text{PosEmb}(i)$$

## 4.3 Self-Attention with Relative XML Paths

The local relation within each web page is essential to the zero-shot relation extraction since the related nodes are more likely to be close in the DOM Tree. As shown in Figure 4, they present a long common

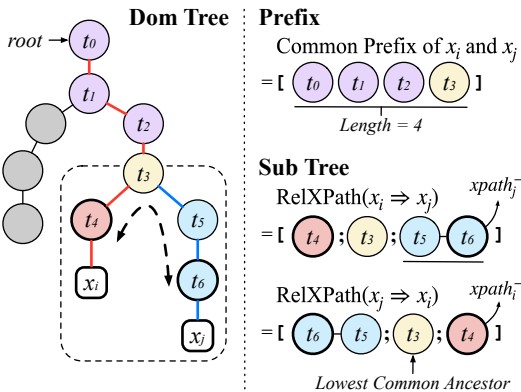

Figure 4: The relative XML Path illustration. In Prefix, we focus on the length of the common prefix of the pair of nodes showing their depth in the DOM Tree, and embed it in the first $\alpha$ attention layers. In Sub Tree, we focus on the shortest path between the pair of nodes, and embed it in the following $\beta$ attention layers.

prefix in their XML Paths, and the rest parts of their XML Paths compose the relative XML Paths between them. The relative XML Paths can be seen as the shortest path between text nodes in the DOM Tree. Therefore, the relative XML Paths are useful signals and could be well transferred into unseen web pages. Enlightened by Yang et al. (2019); Raffel et al. (2020); Peng et al. (2021, 2022), we model the relative XML Paths as bias terms and incorporate them into the multi-layer self-attention of Transformer. Specifically, we embed the common prefix length in the first $\alpha$ layers of self-attention and embed the relative XML Paths tags in the next $\beta$ layers of self-attention, where $(\alpha + \beta)$ equals to the total number of layers. In the case of Figure 4, we embed the common prefix length 4 as well as the relative XML Paths $[t_4, t_3, t_5, t_6]$.

**Extracting Relative XML Paths**  Given a pair of text nodes, $x_i$ and $x_j$, we first extract the common prefix of their XML Paths which shows the path from the root to the lowest common ancestor of these two nodes in the DOM Tree (e.g. $[t_0, t_1, t_2, t_3]$ in Figure 4). We denote the prefix length as $d_{ij}$. The rest parts in the XML Paths shows the path from the lowest common ancestor to the text node. We denote them as $xpath_i^-$ and $xpath_j^-$ which are the XML Paths without the common prefix (e.g. $[t_5, t_6]$ and $[t_4, ]$ in Figure 4). They also compose the shortest path between these

nodes in the DOM Tree:

$$\text{RelXPath}(x_i \Rightarrow x_j) = [\text{rev}(xpath_i^-); t; xpath_j^-]$$
$$\text{RelXPath}(x_j \Rightarrow x_i) = [\text{rev}(xpath_j^-); t; xpath_i^-]$$

where $\text{rev}(\cdot)$ is to reverse the tag sequence; $t$ is the lowest common ancestor of $x_i$ and $x_j$ (e.g. $t_3$ in Figure 4). In the case of Figure 4, $\text{rev}(xpath_j^-)$ equals $[t_6, t_5]$, the lowest common ancestor is $t_3$, and $xpath_i^-$ equals $[t_4, ]$, so $\text{RelXPath}(x_j \Rightarrow x_i)$ equals $[t_6, t_5, t_3, t_4]$.

**Adding Bias Terms**  In the first $\alpha$ layers of the self-attention, we embed the common prefix length $d_{ij}$ as bias terms. The attention weight between $x_i$ and $x_j$ is computed as

$$A_{ij}^\alpha = \frac{1}{\sqrt{d}}(W^Q e_i)^\top (W^K e_j) + \mathbf{b}^{\text{pre}}(d_{ij})$$

where the common prefix length $d_{ij}$ is a bounded integer and each integer is mapped to a specific bias term by $\mathbf{b}^{\text{pre}}(\cdot)$.

In the next $\beta$ layers of the self-attention, we embed the relative XML Paths as bias terms. Following the absolute XML Path embedding (introduced in Section 4.1), we project the embedding of tags in $\text{RelXPath}(x_i \Rightarrow x_j)$ into bias terms. Specifically, we split the relative XML Path at the lowest common ancestor tag and embed each part separately. When embedding $\text{RelXPath}(x_i \Rightarrow x_j)$, the two sub-sequences of tags are $[\text{rev}(xpath_i^-); t]$ and $[t; xpath_j^-]$.

In the equation, $t_m$ is the lowest common ancestor (e.g. $t_3$ in Figure 4); $[t_1, ..., t_m]$ is the path from $x_i$ to the lowest common ancestor (e.g. $[t_4, t_3]$ in Figure 4); $[t_m, ..., t_n]$ is the path from the lowest common ancestor to $x_j$ (e.g. $[t_3, t_5, t_6]$ in Figure 4). The bias term is as follows,

$$\mathbf{b}^{\text{xpath}}(x_i, x_j) = \mathbf{b}\big(\text{Emb}(t_1) \circ ... \circ \text{Emb}(t_m)\big)$$
$$+ \mathbf{b}'\big(\text{Emb}'(t_m) \circ ... \circ \text{Emb}'(t_n)\big) \in \mathbb{R}$$

where $\circ$ is the operation of vector concatenation; Emb is the embedding function; $\mathbf{b}$ is a linear layer projecting embedding to $\mathbb{R}$. We also use two sets of modules to differentiate the two sub-sequences of tags, $(\mathbf{b}, \text{Emb})$ and $(\mathbf{b}', \text{Emb}')$. Thus, the attention weight between $x_i$ and $x_j$ is computed as

$$A_{ij}^\beta = \frac{1}{\sqrt{d}}(W^Q e_i)^\top (W^K e_j) + \mathbf{b}^{\text{xpath}}(x_i, x_j)$$

## 4.4 Contrastive Learning

We observe the sparsity issues in the relation extraction task, where only a small proportion of nodes are annotated as related pairs so the negative cases are much more than the positive ones. To tackle this issue, we adopt the contrastive learning and conduct negative sampling to control the ratio between the positive cases and negative ones.

**Negative Sampling** The number of positive cases and negative cases in the sampling should follow,

$$\#\text{Pos} + \#\text{Neg} = \eta \,; \quad \#\text{Pos} : \#\text{Neg} = \mu$$

where we denote the number of related pairs in the groundtruth as #Pos and the number of negative samples as #Neg; $\eta$ and $\mu$ are two hyperparameters.

**Loss Function** To distinguish the positive samples from the negative ones, we train our model with cross-entropy loss. First, we define the probability of a related pair, $(x_i \rightarrow x_j)$ using the Biaffine attention (Nguyen and Verspoor, 2019) and the sigmoid function $\sigma$.

$$\text{Biaffine}(u, v) = u^\top M v + W(u \circ v) + b$$
$$\mathcal{P}(x_i \rightarrow x_j) = \sigma(\text{Biaffine}(h_i, h_j))$$

where $h_i$ and $h_j$ are the hidden states from ReXMiner corresponding to $x_i$ and $x_j$; $M, W, b$ are trainable parameters; $\circ$ is the vector concatenation. During training, we reduce the cross entropy of training samples against the labels.

$$\mathcal{L} = \sum_{(x_i, x_j)} \text{CrossEntropy}(\mathcal{P}(x_i \rightarrow x_j), L(x_i, x_j))$$

where $L(x_i, x_j)$ is the label of $(x_i, x_j)$, either positive or negative, indicating whether these two nodes are related or not.

## 5 Experiments

We conduct experiments and ablation study of zero-shot relation extraction on the websites of different verticals from the SWDE dataset following the problem settings proposed in Lockard et al. (2020).

### 5.1 Datasets

Our experiments are conducted on the SWDE dataset (Hao et al., 2011). As shown in Figure 2, the SWDE dataset includes websites of three

| Vertical | # Websites | # Web Pages | # Pairs per Web Page |
|---|---|---|---|
| Movie | 8 | 16000 | 34.80 |
| NBA | 8 | 3551 | 11.94 |
| University | 5 | 8090 | 28.44 |

Table 1: The statistics of the SWDE datset.

different verticals, Movie, NBA, and University, and each vertical includes websites of the corresponding topic. For example, `http://imdb.com` and `http://rottentomatoes.com` are collected in the Movie vertical, and `http://espn.go.com` and `http://nba.com` are collected in the NBA vertical. Then the SWDE dataset collects web pages in each website and extracts their HTML source code for web mining tasks. Based on the original SWDE dataset, Lockard et al. (2019, 2020) further annotates the related pairs in the web pages, and propose the zero-shot relation extraction task in web mining. The statistics of the SWDE dataset is shown in Table 1, where we report the total number of websites in each vertical, the total number of web pages in each vertical, and the average number of annotated pairs in each web page.

### 5.2 Experiment Setups

The zero-shot relation extraction task requires that the unseen web pages in the testing set and the existing web pages in the training set are of different verticals. Therefore, we follow the problem settings, and design three tasks based on the SWDE dataset, where we train our model on web pages from two of the three verticals and test our model on the third one. We denote the three tasks as,

- *Movie+NBA⇒Univ*: Train models with the Movie and NBA verticals, and test them on the University vertical;
- *NBA+Univ⇒Movie*: Train models with the NBA and University verticals, and test them on the Movie vertical;
- *Univ+Movie⇒NBA*: Train models with the University and Movie verticals, and test them on the NBA vertical.

We report the precision, recall, and F-1 score.

### 5.3 Compared Methods

We evaluate ReXMiner against several baselines.

**Colon Baseline** The Colon Baseline is a heuristic method proposed in Lockard et al. (2020). It identifies all text nodes ending with a colon (":")

| Model | Unseen Vertical | | | | | | | | | Average |
| | Movie | | | NBA | | | University | | | |
| | Pre | Rec | F1 | Pre | Rec | F1 | Pre | Rec | F1 | F1 |
|---|---|---|---|---|---|---|---|---|---|---|
| Colon[†] | 47 | 19 | 27 | 51 | 33 | 40 | 46 | 31 | 37 | 35 |
| ZSCeres-FFNN[†] | 42 | 38 | 40 | 44 | 46 | 45 | 50 | 45 | 48 | 44 |
| ZSCeres-GNN[†] | 43 | 42 | 42 | 48 | 49 | 48 | 49 | 45 | 47 | 46 |
| MarkupLM[‡] | **48.93** | 40.56 | 44.35 | 44.45 | **71.35** | 54.78 | 58.50 | **62.37** | 60.37 | 53.17 |
| Ours | 45.36 | **49.36** | **47.28** | **65.86** | 64.94 | **65.40** | **68.43** | 60.97 | **64.48** | **59.05** |

Table 2: The experiment results of ReXMiner and baseline models. [†] The results of Colon Baseline and ZeroshotCeres (ZSCeres) are from Lockard et al. (2020). [‡] We introduce the contrastive learning module of ReXMiner to the MarkupLM framework to solve the relation extraction task.

as the relation strings and extracts the closest text node to the right or below as the object. The Colon Baseline needs no training data, so it satisfies the requirement of the zero-shot relation extraction.

**ZeroshotCeres** ZeroshotCeres (Lockard et al., 2020) is a graph neural network-based approach that learns the rich representation for text nodes and predicts the relationships between them. It first extracts the visual features of text nodes from the coordinates and font sizes, and the textual features by inputting the text into a pre-trained BERT model (Kenton and Toutanova, 2019). Then the features are fed into a graph attention network (GAT) (Veličković et al., 2018), where the graph is built based on the location of text nodes in the rendered web page to capture the layout relationships. The relation between text nodes is predicted as a binary classification on their feature concatenation.

**MarkupLM** MarkupLM (Li et al., 2022b) is a pre-trained transformer framework that jointly models text and HTML/XML markup language in web pages. It embeds absolute XML Paths in the embedding layer of the BERT framework and proposes new pre-training tasks to learn the correlation between text and markup language. These tasks include matching the title with the web page, predicting the location of text nodes in the DOM Tree, and predicting the masked word in the input sequence. We use MarkupLM as a backbone model and append it with the contrastive learning module of ReXMiner to solve relation extraction task.

### 5.4 Experimental Results

We report the performance of ReXMiner in Table 2 and compare it with baseline models. From the result, we can see that our proposed model, ReXMiner, achieves the state-of-the-art perfor-

mance in the zero-shot relation extraction task in all three verticals of the SWDE dataset. Specifically, ReXMiner surpasses the second-best model, MarkupLM, by 5.88 in the average F-1 score. In each task, we can observe a remarkable improvement of 2.93, 10.62 and 4.11 in F-1 score when the Movie, NBA, or University verticals are considered as the unseen vertical, respectively.

ZeroshotCeres is the previous state-of-art model proposed to solve the zero-shot relation extraction which leverages the graph neural network to model the structural information. We copy its performance from Lockard et al. (2020). In the comparison with MarkupLM and ReXMiner, we observe that directly modeling the XML Path information using Transformer framework achieves better performance, where MarkupLM and ReXMiner surpass ZeroshotCeres by 7.17 and 13.05 in average F-1 score. The multimodal attention mechanism with absolute XML Path embedding from MarkupLM enhance the performance in each task, and ReXMiner achieves the state-of-the-art overall performance after incorporating the relative XML Paths and the popularity of text nodes.

Though the performance of ReXMiner varies in different verticals, we can safely come to the conclusion that our proposed model, ReXMiner, is superior to the baselines in solving zero-shot relation extraction task in web mining. Further analysis is conducted in Ablation Study and Case Study to study the multimodal features.

### 5.5 Ablation Study

In the ablation study, we aim at studying the role of multimodal features proposed in ReXMiner, including the Relative XML Path Attention and the Popularity Embedding. We introduce three ablation versions of ReXMiner by removing certain

| Model | Unseen Vertical | | | | | | | | | Average |
| | Movie | | | NBA | | | University | | | |
| | Pre | Rec | F1 | Pre | Rec | F1 | Pre | Rec | F1 | F1 |
|---|---|---|---|---|---|---|---|---|---|---|
| ReXMiner | 45.36 | 49.36 | 47.28 | 65.86 | 64.94 | 65.40 | 68.43 | 60.97 | 64.48 | 59.05 |
| - *w/o RelXPath* | 45.60 | 45.68 | 45.64 | 47.13 | 73.54 | 57.44 | 54.82 | 74.63 | 63.21 | 55.43 |
| - *w/o RelXPath + PopEmb* | 48.93 | 40.56 | 44.35 | 44.45 | 71.35 | 54.78 | 58.50 | 62.37 | 60.37 | 53.17 |

Table 3: The results of ablation study, where we compare ReXMiner with two ablation variants, ReXMiner w/o RelXPath and ReXMiner w/o RelXPath + PopEmb. PopEmb denotes the popularity embedding, and RelXPath denotes the relative XPath bias terms.

**NBA+Univ ⇒ Movie** (Prediction result on *Quiz Show.html* )

| Relative XML Path Pattern | Model | Extracted Pairs | |
| | | *Ture Positive* | *False Positive* |
|---|---|---|---|
| RelXPath($x_i \Rightarrow x_j$) = [ [span]; [div]; [ul][li][a] ] | **ReXMiner** *(w/ RelXPath + PopEmb)* | (Color type, Technicolor prints); (Moods, Food for Thought); (Set In, 1958); (Genres, Drama); (Sound by, Dolby); *(Produced by, Buena Vista); (From book, Remembering America); (Keywords, Advertising); (Types, Docudrama)* | (Director, Americana); (Types, Drama); (MPAA Rating, USA); (Keywords, Scandal) |
| | **ReXMiner** *(w/o RelXPath, w/ PopEmb)* | (Color type, Technicolor prints); (Moods, Food for Thought); *(Genres, Drama); (Sound by, Dolby); (From book, Remembering America);* | (Director, Americana); (Genres, Dolby); (Moods, Technicolor prints); (Types, Drama); (Tones, Technicolor prints) |
| | **ReXMiner** *(w/o RelXPath + PopEmb)* | (Color type, Technicolor prints); (Moods, Food for Thought) | (Director, Drama); (Flags, Americana) |

Table 4: The extraction results of the ablation models on `Quiz Show.html` in *NBA+Univ⇒Movie*. The green pairs denote the new true positive predictions compared with the previous ablation model, and the red pairs denote the missing true positive predictions compared with the previous ablation model.

features in Table 3. From Table 3, we compare the performance of all ablation models. We find that using the Popularity Embedding enhances F-1 score by 2.84 and 2.66 in *Movie+NBA⇒Univ* task and *Univ+Movie⇒NBA* task, respectively. After incorporating the Relative XML Path Attention, the F-1 score are further improved in all three tasks. Thus, the ablation model with all multimodal features achieve the highest F-1 score. We conclude that the Relative XML Path contributes to the high precision while the popularity embedding enhances recall leading to the best performance in F-1 score.

## 5.6 Case Study

In Table 4, we show the extraction results of the ablation models on `Quiz Show.html` in *NBA+Univ⇒Movie*. We select one relative XML Path pattern, [span;div;ul,li,a] and list the corresponding extracted pairs into two groups, true positive extractions and false positive extractions. From the results, we can see that ReXMiner with all proposed features shows the best performance, which is also demonstrated in the ablation study. Specifically, by incorporating the Popularity Em-

bedding, ReXMiner (w/o RelXPath, w/ PopEmb) depends on the frequency when predicting the related pairs so it intends to extract more text node pairs and contributes to a higher recall. After adding the Relative XML Path Attention, the extracted pairs are further filtered by the relative XML Path patterns in ReXMiner (w/ RelXPath + PopEmb) so it can extract similar number of true positive pairs and largely reduce the number of false positive cases, but it leads to the missing extraction of *(From book, Remembering America)*.

## 6 Conclusion and Future Work

In this paper, we present ReXMiner, a web mining model to solve the zero-shot relation extraction task from semi-structured web pages. It benefits from the proposed features, the relative XML Paths extracted from the DOM Tree and the popularity of text nodes among web pages from the same website. Specifically, based on MarkupLM, and we further incorporate the relative XML Paths into the attention layers of Transformer framework as bias terms and embed the popularity of text nodes in the embedding layer. To solve the relation ex-

traction task, we append the backbone model with the contrastive learning module and use the negative sampling to solve the sparsity issue of the annotation. In this way, ReXMiner can transfer the knowledge learned from the existing web pages to the unseen ones and extract the related pairs from the unseen web pages. Experiments demonstrate that our method can achieve the state-of-the-art performance compared with the strong baselines.

For future work, we plan to explore the new problem settings with limited supervision, such as few-shot learning and distant supervision, and further study the topological structure information in the DOM Tree to explore more meaningful signals in understanding the semi-structured web pages in web mining tasks.

## 7 Limitations

We build ReXMiner based on MarkupLM and incorporate new features, including the relative XML Paths, the popularity of text nodes, and the contrastive learning. After initializing our model with the pre-trained weights of MarkupLM, the additional modules are finetuned on the datasets of downstream tasks without large-scale pre-training, due to the limited computing resource. We believe more promising results can be achieved if it is possible to pre-train our proposed framework enabling all parameters to be well converged.

## 8 Ethics Statement

Our work focus on the relation extraction in web mining under zero-shot settings. We build our framework using Transformers repository by Huggingface (Wolf et al., 2019), and conduct experiments on the SWDE datasets (Hao et al., 2011). All resources involved in our paper are from open source and widely used in academia. We also plan to release our code publicly. Thus, we do not anticipate any ethical concerns.

### Acknowledgments

We want to thank the anonymous reviewers for their insightful comments. Our work is sponsored in part by NSF CAREER Award 2239440, NSF Proto-OKN Award 2333790, NIH Bridge2AI Center Program under award 1U54HG012510-01, Cisco-UCSD Sponsored Research Project, as well as generous gifts from Google, Adobe, and Teradata. Any opinions, findings, and conclusions or recommendations expressed herein are those of the authors and should not be interpreted as necessarily representing the views, either expressed or implied, of the U.S. Government. The U.S. Government is authorized to reproduce and distribute reprints for government purposes not withstanding any copyright annotation hereon.

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

## Appendix

## A Implementation Details

We use the open-source Transformers framework from Huggingface (Wolf et al., 2019) and build ReXMiner on the base of MarkupLM (Li et al., 2022b). We initialize ReXMiner with the pre-trained weights of MarkupLM, initialize the extra modules with Xavier Initialization (Glorot and Bengio, 2010), and further finetune ReXMiner on the relation extraction tasks. We do not incorporate further pre-training on extra corpus. We use one NVIDIA A6000 to train the model with batch size of 16. We optimize the model with AdamW optimizer (Loshchilov and Hutter, 2018), and the learning rate is $2 \times 10^{-5}$. We set the number of popularity embeddings ($\tau$) as 20, the number of attention layers with the common prefix length ($\alpha$) as 12, the number of attention layers with the relative XML Path ($\beta$) as 3, the total number of samples ($\eta$) as 100, and the ratio between the positive and negative samples ($\mu$) as $\frac{1}{5}$.

| Model | Movie | | |
|---|---|---|---|
| | Pre | Rec | F1 |
| ZSCeres-FFNN[†] | 37 | 50 | 45 |
| ZSCeres-GNN[†] | 49 | 51 | 50 |
| MarkupLM[‡] | **55.98** | 71.30 | 62.72 |
| Ours | 55.15 | **79.50** | **65.12** |

| Model | NBA | | |
|---|---|---|---|
| | Pre | Rec | F1 |
| ZSCeres-FFNN[†] | 35 | 49 | 41 |
| ZSCeres-GNN[†] | 47 | 39 | 42 |
| MarkupLM[‡] | 46.49 | **73.66** | 57.00 |
| Ours | **68.06** | 62.56 | **65.19** |

| Model | University | | |
|---|---|---|---|
| | Pre | Rec | F1 |
| ZSCeres-FFNN[†] | 47 | 59 | 52 |
| ZSCeres-GNN[†] | 50 | 49 | 50 |
| MarkupLM[‡] | 67.09 | **70.56** | 68.78 |
| Ours | **73.08** | 68.73 | **70.84** |

Table 5: The experiment results of ReXMiner and baseline models. [†] The results of ZeroshotCeres (ZSCeres) are from Lockard et al. (2020). [‡] We introduce the contrastive learning module of ReXMiner to the MarkupLM framework to solve the relation extraction task.

## B Zero-shot Relation Extraction on Unseen Websites

In this paper, we propose ReXMiner to solve the zero-shot relation extraction task where web pages in the training set and the testing set are from different verticals. Here, we report the results of the additional experiments for the zero-shot relation extraction on unseen websites. To be more specific, in these additional experiments, the web pages in the training set and the testing set are from the same vertical but different websites. For each vertical in the SWDE dataset, we select a subset of websites as the testing set and train the model with the rest websites. We select "rottentomatoes" and "yahoo" from Movie Vertical, "yahoo" from NBA Vertical, and "ecampustours" and "usnews" from University Vertical. We report the results in Table 5.