# OpenReview forum: "Towards Zero-shot Relation Extraction in Web Mining: A Multimodal Approach with Relative XML Path"
_EMNLP/2023/Conference — EMNLP 2023 Findings_

### Official Review · Reviewer_tb1s · 2023-08-04

**Soundness:** 3

**Excitement:**

3: Ambivalent: It has merits (e.g., it reports state-of-the-art results, the idea is nice), but there are key weaknesses (e.g., it describes incremental work), and it can significantly benefit from another round of revision. However, I won't object to accepting it if my co-reviewers champion it.

**Missing References:**

IMO, the following paper should be cited and discussed. Although their experiments with the SWDE dataset are not comparable due to different settings, the main contribution of the paper is multimodal (text + XML structure) representation learning, where various structural features are encoded along with text elements:

Deng, Xiang, Prashant Shiralkar, Colin Lockard, Binxuan Huang, and Huan Sun. ‘DOM-LM: Learning Generalizable Representations for HTML Documents’. arXiv, 25 January 2022. [https://doi.org/10.48550/arXiv.2201.10608](https://doi.org/10.48550/arXiv.2201.10608).

**Paper Topic And Main Contributions:**

The paper addresses the problem of (untyped) relation extraction (i.e. extracted pairs are not labeled) from web pages and in a zero-shot setting (i.e. testing is done on web pages from a vertical/domain not seen during training). The difficulty is obviously in the generalization to documents that can be different not only in their textual and semantic content, but also in their HTML/XML structure and layout.

The proposed solution follows existing multimodal methods that embed both text and document structure information. More specifically, and as stated in the paper, it is strongly inspired by the MarkupLM model (Li et al., 2021), which embeds absolute XML Paths along with textual elements within a transformer encoder.

The main contribution of the paper is the extension of the MarkupLM model (Li et al., 2021) with 2 features: popularity of text nodes (i.e. number of web pages where a text element occurs) and relative XML Path between text elements. The first feature is encoded in the input sequence embeddings (of text nodes), and the second is encoded as a bias term in the self-attention components. The experiments and ablation study show that the two features improve performance.

**Questions For The Authors:**

A. There seems to be a confusion in the use of the notions of text nodes, words and tokens. A text node is an element from the DOM tree and can be a token or word, a phrase, a sentence or even a paragraph. Could you please clarify what is the nature of the elements in the input sequence for the model, i.e. what w_i means in the (w_i, xpath_i) notation? Is it a text node or a token? If they are tokens, are the text nodes tokenized with a BPE tokenizer?

B. If the input of the model is a sequence of tokens (with their xpaths), then h_i and h_j in the Biafine equation (line 383) are representations of tokens, and x_i and x_j (line 384) are tokens, not text nodes. So how does the system group tokens to produce pairs of text nodes, such as "(Color type, Technicolor prints)" ?

C. It is claimed that "contrastive learning" is used to train the model (section 4.4) but I am not sure if that's correct: the loss as defined in lines 377-393 seems to be a normal supervised cross-entropy loss (prediction vs true label on a single sample) and not a contrastive loss.  So it seems that the negative sampling is used simply to alleviate the problem of class imbalance and not for a contrastive loss. Could you please clarify?

**Reasons To Accept:**

- The proposed extension of the MarkupLM model, with the encoding of text node "popularity" and relative XML paths between text nodes, is relatively simple and improves performance in zero-shot relation extraction setting, as shown by the experiments and the ablation study.

**Reasons To Reject:**

- Two important technical aspects of the proposed method are missing or are not clear:
	- (1) Regarding the representation of model input: from section 4 and in particular lines 280-284, the model input is defined as the sequence of "text nodes" (w_i, xpath_i) from the XML/HTML document, which is confusing, as text nodes can be phrases, sentences or even paragraphs... In Figure 3 though, it seems that the model input is a sequence of tokens, not text nodes, which would make more sense, but the paper does not mention tokenization at all.
	- (2) Regarding the extractions: from the provided examples (Table 4), the model extracts pairs of text nodes, such as "(Color type, Technicolor prints)". If, as it is probably the case, model input is a sequence of tokens (from tokenized text nodes), then h_i and h_j in the Biaffine component (equation at line 383) are representations of tokens, not text nodes and hence, P(x_i, x_j) is about tokens x_i and x_j. Therefore, the paper lacks a description of how tokens are grouped in order to produce pairs of text nodes from the predicted pairs of tokens.

**Reproducibility:**

4: Could mostly reproduce the results, but there may be some variation because of sample variance or minor variations in their interpretation of the protocol or method.

**Reviewer Confidence:**

4: Quite sure. I tried to check the important points carefully. It's unlikely, though conceivable, that I missed something that should affect my ratings.

**Typos Grammar Style And Presentation Improvements:**

Typos:

line 143: gathering --> gather

figure 2 caption: hundreds web pages --> hundreds of web pages

Throughout the paper: the definite article "the" is sometimes used in plural noun phrases where it should not be, e.g. "which rely on the syntactic constraints" --> "which rely on syntactic constraints". Proofreading is probably needed to fix all the cases.

---

> ### Author Rebuttal · Authors · 2023-08-29
>
> We appreciate the reviewer for the detailed comments and insightful suggestions. Here we would clarify the confusing points in our paper and will also revise the related parts in our future version. All the typos, missing references, and presentation issues will also be well handled.
>
> **Clarification of the model inputs**
>
> To encode the textual contents of a web page into a language model, we follow the previous works [1][2] where the contents in all the nodes are concatenated as a long string to be fed into the model. We take note of the beginning and ending indices to reconstruct each node. As for the XML paths, all the words in a given node share the same XML path encoding. So, given a node in web page, $([w_1,...,w_n], xpath)$, we tokenize it into such input sequence: $(w_1, xpath), (w_2, xpath),..., (w_n, xpath)$.
>
> **Clarification of the extraction step**
>
> As we discussed, when converting the nodes into the input sequence, we take note of the beginning and ending indices. Our task is to extract the related pairs of nodes, so we extract the hidden states corresponding to the end index as the representation for the given node. These hidden states are fed into the Biaffine layers to further predict the relationship between nodes.
>
> **Ambiguous use of contrastive loss**
>
> Thanks for pointing out this issue. We will remove the ambiguous use of the term and make the narration clearer.
>
> - [1] MarkupLM: Pre-training of Text and Markup Language for Visually-rich Document Understanding
> - [2] DOM-LM: Learning Generalizable Representations for HTML Documents

---

### Official Review · Reviewer_3bEb · 2023-08-05

**Soundness:** 3

**Excitement:**

4: Strong: This paper deepens the understanding of some phenomenon or lowers the barriers to an existing research direction.

**Paper Topic And Main Contributions:**

This paper studies the approach to understand semi-structured web pages utilizing the relation between text nodes within and across pages.
It proposes a framework ReXMiner to encode the shortest relative paths in document object model and the popularity info of text node.
It also uses the contrastive learning approach to facilitate the training.
Experiments conducted on SWDE dataset shows the model achieves or outperforms the SOTA performance.


**Questions For The Authors:**

I am wondering if the author has considered IDF as the popularity of text node has been considered.


**Reasons To Accept:**

Paper is well structured and clear to follow.
Paper tackles one important problem in web text mining field.
Paper shows the proposed solution can outperform the state of art models.
Paper show cases the gained benefits by combining more structural or popularity information.


**Reasons To Reject:**

Authors can consider to test more on other datasets.
It will be even better to compare the inference/training time.



**Reproducibility:**

4: Could mostly reproduce the results, but there may be some variation because of sample variance or minor variations in their interpretation of the protocol or method.

**Reviewer Confidence:**

3: Pretty sure, but there's a chance I missed something. Although I have a good feel for this area in general, I did not carefully check the paper's details, e.g., the math, experimental design, or novelty.

---

> ### Author Rebuttal · Authors · 2023-08-29
>
> We appreciate the reviewer's recognition of the effectiveness of our framework. Here we provide more experiments and results to further demonstrate our proposed method.
>
> **More experiments and results**
>
> Our motivation is to leverage the relative information of the XML path to help language models better understand the structure of web pages especially when handling unknown web pages.
>
> In our current problem setting, we follow the task definition in [1] and explore the hardest setting where the models need to transfer the knowledge from the existing vertical to the unknown vertical.
>
> We add one more problem setting where we ask the model to extract related pairs on the unknown websites. To be more specific, the training set and testing set are of the same vertical but from different websites. We believe this new setting is slightly easier than our current setting since websites of the same vertical may share common domain knowledge. Here are the results on the Movie vertical (Due to time constraints, we only finished this one setting. We will report the other two verticals in our final version).
> |                | Precision | Recall | F1-Score |
> |----------------|-----------|--------|----------|
> | ZeroshotCeres  | 49.0      | 51.0   | 50.0     |
> | MarkupLM       | 55.98     | 71.30  | 62.72    |
> | ReXMiner(Ours) | 55.15     | 79.50  | 65.12    |
>
> More details about the problem settings defined in [1] (from Table 1 of [1]).
>
> | Setting Level | Site-specific | Test set's relation to Training set     | Example                                                                      |
> |---------------|---------------|-----------------------------------------|------------------------------------------------------------------------------|
> | III           | Yes           | Different web pages of the same website | Train with amazon.com/001.html and test with amazon.com/002.html             |
> | II (Added new setting)           | No            | Different websites of the same vertical | Train with amazon.com/001.html and test with <another_shopping>.com/001.html |
> | I (Our current setting)      | No            | Different verticals                     | Train with amazon.com/001.html and test with NBA.com/001.html                |
>
> **Use of IDF as the popularity of text node**
>
> We thank the reviewer's insightful suggestions. Indeed, our definition of the node's popularity is quite related to the well-known word IDF. In our definition of popularity, we count the number of web pages where the given text node exists and embed it through the equation in Line 276: $\lfloor \tau \log(pop(w)) / \log(N) \rfloor$. This definition is similar to the word IDF but uses a different way to normalize the value. Since the word IDF can range from 0 to +inf, which makes it hard to embed the values into vectors. We believe that our method produces bounded values and contains the same information as IDF.
>
> - [1] ZeroShotCeres: Zero-Shot Relation Extraction from Semi-Structured Webpages

---

### Official Review · Reviewer_KQqa · 2023-08-12

**Soundness:** 4

**Excitement:**

3: Ambivalent: It has merits (e.g., it reports state-of-the-art results, the idea is nice), but there are key weaknesses (e.g., it describes incremental work), and it can significantly benefit from another round of revision. However, I won't object to accepting it if my co-reviewers champion it.

**Paper Topic And Main Contributions:**

In this paper, a new method called ReXMiner is proposed for zero-shot relation extraction in web mining. It encodes the shortest relative paths in the DOM tree of the web page and counts the occurrence of the same text node across different web pages. Experiments on public benchmarks show that this method outperforms the state-of-the-art baselines in the task of zero-shot relation extraction in web mining.

**Questions For The Authors:**

A. The relative path sequences seem to be simply put together so information about direction and lowest common ancestor is not fed into the model. Does this information have any effect on the model?

B. In this model, path prefixes are added to the first several layers of attention weight as bias and relative paths are added to the next layers of attention weight as bias. How do they interact with each other and ultimately affect the model?

C. The number of these layers is controlled by hyperparameters, how are these hyperparameters determined and do they have a significant effect on the model?

**Reasons To Accept:**

This paper proposes a kind of novel and effective feature fomulation for web pages by encoding the shortest relative paths in the DOM tree and counting the occurrence of the text node.

The transformer-based model ReXMiner using the above features reaches the state-of-the-art in the task of zero-shot relation extraction for web mining, which is hopeful to facilitate the exploitation of the unlabelled web corpus.

**Reasons To Reject:**

The contribution of this paper seems somewhat insufficient as a long paper. As a considerable part of ReXMiner comes from the existing work of the model MarkupLM, the adding part including relative path bias and popularity embeddings are more like an incremental work. I believe this paper should be enriched in breadth or depth to fully demonstrate the advantages and importance of the new approach. For example, more experiments on other datasets and tasks about web pages, like what MarkupLM does, or more analytical experiments illustrating the principles of the new method.

**Reproducibility:**

3: Could reproduce the results with some difficulty. The settings of parameters are underspecified or subjectively determined; the training/evaluation data are not widely available.

**Reviewer Confidence:**

4: Quite sure. I tried to check the important points carefully. It's unlikely, though conceivable, that I missed something that should affect my ratings.

---

> ### Author Rebuttal · Authors · 2023-08-29
>
> We thank the reviewer for the insightful comments and suggestions. We will clarify the confusing points and add them to our later version.
>
> **Experiments on other datasets and analysis**
>
> We understand that it is important to validate a method with more data and experiments. However, through our literature review, the SWDE with extended relation labels [1] is the only existing dataset designed for HTML-aware relation extraction in web pages. Most of the existing datasets for relation extraction tasks consist of pure text and skip the rich HTML data even if their data source may be web pages [2][3].
>
> We believe that the current SWDE dataset is extensive enough which includes 3 verticals, 21 websites, and 27641 web pages all in total, and this is also the dataset used by baselines to demonstrate their performance in the relation extraction tasks.
>
> To further evaluate our method, we add more experiment settings. In our current version, we explore the performance of ReXMiner when transferring the knowledge from existing verticals to unknown verticals (e.g. Movie + University → Nbaplayer). Now, we add one more setting according to the task definitions in [4]. We train our model on a set of known websites and apply it to unknown websites. Here the training set and testing set are of the same vertical but different websites. Due to time constraints, Here are the new results of Movie vertical. We will report the other two verticals in our final version.
>
> |                | Precision | Recall | F1-Score |
> |----------------|-----------|--------|----------|
> | ZeroshotCeres  | 49.0      | 51.0   | 50.0     |
> | MarkupLM       | 55.98     | 71.30  | 62.72    |
> | ReXMiner(Ours) | 55.15     | 79.50  | 65.12    |
>
> More details about the problem settings defined in [1] and [4] (from Table 1 of [4]).
>
> | Setting Level | Site-specific | Test set's relation to Training set     | Example                                                                      |
> |---------------|---------------|-----------------------------------------|------------------------------------------------------------------------------|
> | III           | Yes           | Different web pages of the same website | Train with amazon.com/001.html and test with amazon.com/002.html             |
> | II (Added new setting)           | No            | Different websites of the same vertical | Train with amazon.com/001.html and test with <another_shopping>.com/001.html |
> | I (Our current setting)      | No            | Different verticals                     | Train with amazon.com/001.html and test with NBA.com/001.html                |
>
> **Details about the direction of relative XML path and lowest common ancestor**
>
> In short, we indeed encode the direction of XML paths and the lowest common ancestor in our Self-Attention with Relative XML Paths. We understand that the direction of relative XML paths and lowest common ancestor are of great importance to the relationship between text nodes in web mining. We encode them as follows.
>
> 1. *Lowest common ancestor:* As shown in Line 352 and 353, the t_m denotes the lowest common ancestor and its embedding is part of the XML Path bias.
> 2. *The order of XML tags in relative XML paths:* We encode the order of XML tags through the concatenation of embedding vectors. As shown in Line 352 and 353, we first embed each XML tag into a short vector and concatenate these vectors sequentially. Then we use a linear layer to convert the concatenated vector into a bias term. Here the concatenation ensures that the bias term involves the direction information.
> 3. *The direction of relative XML paths:* Similarly, in Line 352 and 353, the relative XML path from node $x_i$ to $x_j$ is divided into two parts, and we embed each part with different linear layers, b, and b$^\prime$, which ensures that the added bias terms of $x_i\to x_j$ and $x_j\to x_i$ are different.
>
> **Details about the Self-Attention with Relative XML Paths**
>
> In the design of Self-Attention with Relative XML Paths, we would like to encode both the depth information and the exact relative XML paths. The depth is a simple scalar and easy to learn as a bias term, while the exact relative XML paths are more informative and computing-intensive to learn as a bias term. We do a trade-off between these two and embed them separately in the multi-layer attention as bias terms. In the first alpha layers, the added terms are from the depth value which is easy to learn and well combined with the pretrained weight. In the next beta layers, the added terms are from the exact relative XML paths and the hidden states already contain rich semantic information, so the combined signals can be well kept during the final prediction.
>
> **Hyperparameters of the Self-Attention with Relative XML Paths**
>
> As discussed, we would like to encode both the depth information and the exact relative XML paths into the attention layers, so we design two different bias terms. To balance the computing, we use more layers to encode the depth information and the last 3 layers to encode the exact relative XML paths. We believe the choice of hyperparameters is not a big issue in our framework and other combinations might lead to a better result.
>
> - [1] OpenCeres: When Open Information Extraction Meets the Semi-Structured Web
> - [2] REDTab: A Relation Extraction Dataset for Knowledge Extraction from Web Tables
> - [3] DocRED: A Large-Scale Document-Level Relation Extraction Dataset
> - [4] ZeroShotCeres: Zero-Shot Relation Extraction from Semi-Structured Webpages

---

### Official Review · Reviewer_uVRd · 2023-08-15

**Soundness:** 4
**Typos Grammar Style And Presentation Improvements:** Too many arxiv references, some of wh…

**Excitement:**

3: Ambivalent: It has merits (e.g., it reports state-of-the-art results, the idea is nice), but there are key weaknesses (e.g., it describes incremental work), and it can significantly benefit from another round of revision. However, I won't object to accepting it if my co-reviewers champion it.

**Missing References:**

[1] Li, Zimeng, et al. "WIERT: Web Information Extraction via Render Tree." Proceedings of the AAAI Conference on Artificial Intelligence. Vol. 37. No. 11. 2023.

[2] Xie, Chenhao, et al. "Webke: Knowledge extraction from semi-structured web with pre-trained markup language model." Proceedings of the 30th ACM International Conference on Information & Knowledge Management. 2021.

**Paper Topic And Main Contributions:**

This paper addresses the challenge of understanding semi-structured web pages in web mining, particularly when little is known about the subject or template of a new page. The authors note that current methods, which involve embedding XML source code into the transformer or encoding the rendered layout with graph neural networks, do not adequately account for the relationships between text nodes within and across pages.

The main contributions of this paper are the development and implementation of ReXMiner, which offers a more accurate and efficient method for key-value pair extraction within a web page. ReXMiner achieves this by encoding the shortest relative paths in the Document Object Model (DOM) tree of the web page. It also incorporates the popularity of each text node by counting the occurrence of the same text node across different web pages.
The authors use contrastive learning to address the issue of sparsity in relation extraction.

**Questions For The Authors:**

1. What are the zero-shot results of recent LLMs, e.g., gpt4, on the task?
2. This point may worth discussion: in industrial practice, few shot may always be acceptable, so why do authors insist on zero-shot as the paper position?

**Reasons To Accept:**

1. easy to follow
2. The technical details are solid and reasonable.

**Reasons To Reject:**

1. lack comparison to existing PLM-based methods, such as [1,2] and openai-gpt based methods.
2. The components in the paper are popular approaches, I suggest that the authors give more insight and example on how each component (e.g. contrasive loss) works on each scenario.
3. poor reproducibility, without any code available for evaluation.






References (see missing references)

**Reproducibility:**

2: Would be hard pressed to reproduce the results. The contribution depends on data that are simply not available outside the author's institution or consortium; not enough details are provided.

**Reviewer Confidence:**

4: Quite sure. I tried to check the important points carefully. It's unlikely, though conceivable, that I missed something that should affect my ratings.

---

> ### Author Rebuttal · Authors · 2023-08-29
>
> **Reproduction**
>
> We believe it is essential to open-source our code to reproduce the result and benefit future research. As claimed in the last paragraph of the Introduction Section, we will open-source the code on GitHub. We cannot open-source it now simply because of the anonymity requirements from EMNLP.
>
> **The Motivation of Zero-shot WebIE**
>
> Zero-shot web mining is an important task in both academia and industry. It has been pursued for years and has significant industry applications [1,2,3,4]. It benefits future research and applications since online services need to be timely updated and adapted to the new changes on the internet but it is not easy to get either few-shot examples or updated models in a short time.
>
> Due to the fast pace of internet development, it is not always possible to get timely updates on the current model. Although it might be true that few-shot examples can be easily obtained in industrial practice, it also takes money and time to update the ongoing model with the few-shot data, especially when it comes to small businesses or even individual users. Therefore, it is worthwhile to explore web mining algorithms or frameworks that can be well generalized to unknown data in a zero-shot setting.
>
> **Experiment Result with OpenAI LLMs**
>
> We also observe the extraordinary performance of current LLMs in zero-shot and few-shot learning and we believe our focus of this paper is orthogonal to the development of LLMs.
>
> LLMs are good at understanding natural language and solving general NLU tasks thanks to the extensive pre-training data. However, in our task, we focus on a specific task: the relation extraction in web pages. We seek signals from both the XML paths and the textual contents on the web page. We design special modules for transformer architecture to let LMs better leverage the XML paths. We believe that our work shows the importance of relative relation between HTML nodes in web mining.
>
> We believe the HTML data should be already included in the pre-training of LLMs from OpenAI and assume the HTML source code is presented in the textual format or rendered as images. However, it is still an open question of how to leverage the structural signals from HTML in a fine-grained way as what we can realize in the era of BERT-like models.
>
> Furthermore, if LLMs are used to solve the web mining problem, we believe that the challenge is to design proper prompt templates and select suitable demo samples to let LLMs understand the HTML structure in the pure text format. Due to the time constraint,  We ran the experiment with GPT-4 on 200 random samples in Movie vertical and reported the performance here. We input the same information to GPT-4 as our model.
> |                     | Precision | Recall | F1-Score |
> |---------------------|-----------|--------|----------|
> | GPT-4 (200 samples) | 33.60%     | 19.74%  | 24.87%    |
> | ReXMiner (Fulll set)          | 45.36%     | 49.36%  | 47.28%    |
>
> We can observe that although GPT-4 is powerful in zero-shot and few-shot learning, it is still a non-trivial task to learn from XML paths to extract related pairs.
>
> **Experiment Result with WebKE**
>
> We thank the reviewer for pointing out the missing reference. I will cite the paper in our final version. Here we compare the difference between our framework and WebKE and report the performance with the WebKE model on our task. We will include this in our future version.
>
> 1. *Different task settings:* WebKE model is designed for site-specific OpenIE in web pages. As defined in [1] and [2], the site-specific OpenIE depends on the knowledge learned from the web pages of the same website. To be more specific, the training set and testing set of WebKE are from the same website. However, following the zero-shot setting defined in [2], our task is zero-shot OpenIE which is harder than site-specific OpenIE. In our problem setting, the training set and testing set are of different domains. The model is asked to transfer knowledge from one domain to another. So we have three settings: Movie+NBA⇒Univ, NBA+Univ⇒Movie, and Univ+Movie⇒NBA. The three settings defined by [1] and [2] are as follows (from Table 1 of [2]):
>
> | Setting Level | Site-specific | Test set's relation to Training set     | Example                                                                      |
> |---------------|---------------|-----------------------------------------|------------------------------------------------------------------------------|
> | III (WebKE)       | Yes           | Different web pages of the same website | Train with amazon.com/001.html and test with amazon.com/002.html             |
> | II            | No            | Different websites of the same vertical | Train with amazon.com/001.html and test with <another_shopping>.com/001.html |
> | I (Ours)      | No            | Different verticals                     | Train with amazon.com/001.html and test with NBA.com/001.html                |
>
> 2. *Different web page structure encoding:* As discussed in the paper, one motivation of this work is to let language models better understand the structural information of web pages. In WebKE, they render the web pages into an image and use the coordinates of the text nodes in the web page to encode the structural information. In comparison, we skip the rendering and directly encode the XML paths. We believe that our method is more efficient since rendering may take a long time to process, and our method is more robust since the coordinates of the text nodes may be influenced by the different rendering platforms (web pages may look different from cell phone and laptop) but the XML paths are always the same.
>
> 3. *Different solution for negative cases:* Both our work and WebKE have noticed a large number of negative cases in the relation extraction. In our work, we solve this issue through negative sampling and contrastive loss. In WebKE, they extract relations and objects by predicting the start and end of each phrase and also use heuristic filtering to further process the prediction results.
>
> To quantify the difference between WebKE and our method, we run the exact same code of WebKE and only change the training set and testing set to the zero-shot relation extraction setting. Due to the time constraint, we only run the experiment setting of Movie+NBA⇒Univ. The results are:
> |                     | Precision | Recall | F1-Score |
> |---------------------|-----------|--------|----------|
> | WebKE | 0.41%     | 13.59%  | 0.80%    |
>
> We believe the difference is due to the different task settings and how to encode the web page structure. In the WebKE setting, the training set and testing set are different web pages but from the same website. The layout of the web pages can be similar so the coordinates can be easily used to learn the relation and further predict the object. However, in our zero-shot relation extraction setting, the training set and testing set are of different domains, making it hard to extract these pairs through coordinates. The relation revealed by the XML paths in our approach stays the same and contributes to our better performance.
>
> **Each Component in Our Framework**
>
> Here we clarify our insight into designing each module in our framework.
>
> 1. *Relative XML Path:* While it is common to acquire meaningful signals from the relative position of tokens in a sequence, it hasn’t been a feasible way to leverage relative XML paths in web mining. In our work, we observe the long common prefix in the relative node pairs and implement the relative XML path modules. We separate the relative XML information into two parts: the depth and the exact relative path, which serves as a clear encoding of the relative XML paths.
>
> 2. *Popularity Embedding:* The frequency is commonly used to reveal the importance of a phrase in the text. As discussed in the paper, the relation text nodes are more frequent than the other nodes and their neighboring words are different. Therefore, we combine the frequency information into the context embeddings to see the performance.
>
> 3. *Contrastive Learning:* In the task setting of zero-shot relation extraction, the relationship is explicitly extracted. For example, “Age:”, and “24” are two text nodes in the web page, and we need to extract both of them as a related pair. So the task reduces to a binary classification between any two text nodes, which introduces a large proportion of negative cases. To balance the positive and negative cases, we use negative sampling to control the ratio.
>
> - [1] OpenCeres: When Open Information Extraction Meets the Semi-Structured Web
> - [2] ZeroShotCeres: Zero-Shot Relation Extraction from Semi-Structured Webpages
> - [3] CERES: Distantly Supervised Relation Extraction from the Semi-Structured Web
> - [4] Zero-shot Entity Extraction from Web Pages

---

### Meta-Review · Area_Chair_pYVX · 2023-09-07

**Recommendation:** 2

**Metareview:**

The paper presents ReXMiner, a new approach to zero-shot relation extraction from webpages that encodes relative shortest paths in the Document Object Model (DOM).  The sense of zero-shot here is following prior work (e.g. Lockard et. al.) in that the model is trained on one domain (e.g. NBA), and tested on webpages from another domain (e.g. Movies).

Reviewers appreciated the simplicity of the approach as a nice extension to MarkupLM, and the fact that the paper is relatively well written.
 However, some reviewers comments raised concerns about whether the contribution is incremental with respect to prior work to justify acceptance as a long paper.  Reviewers also indicated they would like to see a comparison to a baseline that uses LLMs.

I have some concerns about the experiments, for example, what are the dev/test splits used to select hyperparameters - are these the same that were were used in prior work?  The ZeroShotCeres paper indicates they do not use a development set of pages from the target websites, whereas in Appendix A the paper mentions: "As for the hyper-parameters, we select a subset of web pages from each web site in the SWDE dataset as validation set to find the best hyper-parameters.", which seems to indicate that pages from the target website was included in the development set, so this seems like it may not be a fair comparison.

---

### Decision · Program_Chairs · 2023-10-07

**Decision:**

Accept-Findings

**Comment:**

The paper presents ReXMiner, a new approach to zero-shot relation extraction from webpages that encodes relative shortest paths in the Document Object Model (DOM).  The sense of zero-shot here is following prior work (e.g. Lockard et. al.) in that the model is trained on one domain (e.g. NBA), and tested on webpages from another domain (e.g. Movies).

Reviewers appreciated the simplicity of the approach as a nice extension to MarkupLM, and the fact that the paper is relatively well written.
 However, some reviewers comments raised concerns about whether the contribution is incremental with respect to prior work to justify acceptance as a long paper.  Reviewers also indicated they would like to see a comparison to a baseline that uses LLMs.

I have some concerns about the experiments, for example, what are the dev/test splits used to select hyperparameters - are these the same that were were used in prior work?  The ZeroShotCeres paper indicates they do not use a development set of pages from the target websites, whereas in Appendix A the paper mentions: "As for the hyper-parameters, we select a subset of web pages from each web site in the SWDE dataset as validation set to find the best hyper-parameters.", which seems to indicate that pages from the target website was included in the development set, so this seems like it may not be a fair comparison.